# Not to Rush—Laboratory Parameters and Procedural Complications in Patients Undergoing Left Atrial Appendage Closure

**DOI:** 10.3390/jcm11216548

**Published:** 2022-11-04

**Authors:** David Zweiker, Lukas Fiedler, Gabor G. Toth, Andreas Strouhal, Georg Delle-Karth, Guenter Stix, Harald Gabriel, Ronald K. Binder, Martin Rammer, Michael Pfeffer, Paul Vock, Brigitte Lileg, Clemens Steinwender, Kurt Sihorsch, Florian Hintringer, Silvana Mueller, Fabian Barbieri, Martin Martinek, Wolfgang Tkalec, Nicolas Verheyen, Klemens Ablasser, Andreas Zirlik, Daniel Scherr

**Affiliations:** 1Department of Cardiology, Medical University of Graz, Auenbruggerplatz 15, 8036 Graz, Austria; 2Third Department for Cardiology and Intensive Care, Clinic Ottakring, 1160 Vienna, Austria; 3Department of Internal Medicine, Cardiology and Nephrology, Hospital Wiener Neustadt, 2700 Wiener Neustadt, Austria; 4Department of Cardiology, Hospital Nord–Klinik Floridsdorf, 1210 Vienna, Austria; 5Department of Internal Medicine II, Medical University of Vienna, 1090 Vienna, Austria; 6Department of Internal Medicine II, Klinikum Wels-Grieskirchen, 4600 Wels, Austria; 7Department of Internal Medicine III, University Hospital St. Pölten, 3100 St. Pölten, Austria; 8Department of Cardiology, Kepler University Hospital, 4020 Linz, Austria; 9Department of Internal Medicine III, Medical University of Innsbruck, 6020 Innsbruck, Austria; 10Charité–Universitätsmedizin Berlin, Corporate Member of Freie Universität Berlin and Humboldt-Universität zu Berlin, Department of Cardiology, 10117 Berlin, Germany; 11Department of Internal Medicine II, Elisabethinen Hospital, 4020 Linz, Austria; 12Department of Cardiology, Cardiovascular Research Institute Maastricht (CARIM), Maastricht University Medical Centre, 6229 HX Maastricht, The Netherlands

**Keywords:** left atrial appendage closure, atrial fibrillation, complications, haemoglobin, dialysis

## Abstract

Background: As a preventive procedure, minimizing periprocedural risk is crucially important during left atrial appendage closure (LAAC). Methods: We included consecutive patients receiving LAAC at nine centres and assessed the relationship between baseline characteristics and the acute procedural outcome. Major procedural complications were defined as all complications requiring immediate invasive intervention or causing irreversible damage. Logistic regression was performed and included age and left-ventricular function. Furthermore, the association between acute complications and long-term outcomes was evaluated. Results: A total of 405 consecutive patients with a median age of 75 years (37% female) were included. 47% had a history of stroke. Median CHA2DS2-VASc score was 4 (interquartile range, 3–5) and the median HAS-BLED score was 3 (2–4). Major procedural complications occurred in 7% of cases. Low haemoglobin (OR 0.8, 95% CI 0.65–0.99 per g/dL, *p* = 0.040) and end-stage kidney disease (OR 13.0, CI 2.5–68.5, *p* = 0.002) remained significant in multivariate analysis. Anaemia (haemoglobin < 12 and < 13 g/dL in female and male patients) increased the risk of complications 2.2-fold. Conclusions: The major complication rate was low in this high-risk patient population undergoing LAAC. End-stage kidney disease and low baseline haemoglobin were independently associated with a higher major complication rate.

## 1. Introduction

Left atrial appendage closure (LAAC) is an interventional procedure for stroke prevention in patients with atrial fibrillation (AF). While oral anticoagulation (OAC) is considered standard treatment in most patients with an elevated stroke risk, LAAC may be performed as an alternative in patients with a high thromboembolic risk and are contraindicated to long-term OAC therapy (recommendation IIb by ACC and ESC guidelines) [1,2]. Another emerging indication is recurrent stroke despite adequate antithrombotic treatment [3]. Since the approval of LAAC by authorities, there has been some emerging experience with this technology in the western world [4,5].

While LAAC showed favourable results in patients who possess contraindications to OAC [6,7,8] and even compared to direct OAC (DOAC) therapy [9,10], it still represents a purely preventive procedure without proven acute benefit to the patient. Therefore, the avoidance of procedural complications is important. The identification of predictors for complications may improve the patient selection process and guide optimal procedural preparation.

The goal of this analysis was to identify predictors for acute procedural complications in consecutive patients undergoing LAAC and to evaluate the importance of centre experience on complications and long-term outcomes.

## 2. Materials and Methods

This is a subanalysis of the Austrian LAAC Registry (NCT03409159). The registry includes all patients, who underwent LAAC in Austria between 2010 and 2021. The study is approved by the institutional review board of the Medical University of Graz (29-355 ex 16/17).

### 2.1. Recruitment and Procedure

Details about the recruitment, indications, and outcomes in LAAC patients in Austria have been published elsewhere [11]. In short, patients with AF, a high thromboembolic risk, and either contraindications, intolerances, or ineffectiveness of OAC were evaluated at each of the referral centres in Austria. Both the decision to perform LAAC and the selection of device was left to the operators’ or institutes’ discretion, according to international guidelines [1] and the vendors’ standard operating protocol. Used devices were Watchman™, Watchman FLX™ (Boston Scientific, Marlborough, MA, USA), Amplatzer Cardiac Plug™, Amplatzer Amulet™ (Abbott Laboratories, North Chicago, IL, USA) or LAmbre™ (Lifetech Scientific, Shenzhen, China). Post-procedural management and antithrombotic treatment were left to the operators’ discretion, tailored according to the patients’ individual risk profiles. Follow up included transoesophageal echocardiography and regular ambulatory checks.

### 2.2. Data Collection

Clinical data of consecutive patients undergoing LAAC at participating centres were captured, considering recommendations from the European Heart Rhythm Association (EHRA) and the European Association of Percutaneous Cardiovascular Interventions (EAPCI) [5] by either a local representative or an external reviewer. Mortality data was included from the Austrian government’s population registry. Patient inclusion was complete in all centres until the end of 2019 and in 8/9 centres (88.9%) until the end of 2020.

### 2.3. Endpoints

Major procedural complications were defined as all complications requiring immediate invasive intervention. Cardiac tamponade was defined as pericardial effusion requiring pericardiocentesis or surgery. Access site complications were documented if they required invasive intervention. Ischaemic stroke was defined as clinically relevant ischaemic stroke according to current guidelines [12]. Other procedural complications were summarized as minor procedural complications. Bleeding requiring transfusion was explicitly defined as a minor procedural complication. Pre-procedural anaemia was defined as haemoglobin <12 mg/dL in female patients and <13 mg/dL in male patients [13]. For long-term outcome assessment, one-year survival was analysed.

### 2.4. Statistical Analysis

We expressed parameters as count (proportion), mean ± standard deviation, or median (interquartile range), as appropriate. Depending on the presence of a normal distribution, calculated by the Shapiro-Wilk test, either the student’s *t*-test or the Wilcoxon signed-rank test was used for univariable analysis. For outcome analyses, multivariable analysis was performed using logistics regression analysis, including all parameters with significant differences between patients with and without complications (*p* < 0.05), age, and left ventricular function. For the interdependent variables (e.g., haematocrit and haemoglobin), only one parameter with the highest significance was allowed into the multivariable analysis. Multiple imputation with five iterations was used for missing values. A two-sided *p* value < 0.05 was considered significant. For statistical analysis, R 4.2.0 (The R Project, Vienna, Austria) and RStudio 2022.02.1 Build 461 (RStudio PBC, Boston, MA, USA) were used.

## 3. Results

A total of 405 patients undergoing LAAC between November 2010 and December 2021 at 9 Austrian centres were included into this analysis.

### 3.1. Baseline Characteristics

The median age was 75 years and 37% were female (Table 1). The median (IQR) body mass index was 27 (24–30) kg/m^2^ and body surface area was 1.92 ± 0.21 m^2^. The median (IQR) CHA_2_DS_2_-VASc score was 4 (3–5) and HAS-BLED score was 3 (2–4). The most common comorbidities were arterial hypertension (88.4%), coronary artery disease (41%), and previous stroke (47%). Ischaemic stroke was present in 25%, haemorrhagic stroke in 26% and anaemia in 38%. Other comorbidities are summarized in Table 1.

The median haemoglobin was 12.5 g/dL and median creatinine was 1.1 mg/dL, leading to a median estimated glomerular filtration rate of 75 mL/min/1.73 m^2^.

### 3.2. Procedural Details and Outcome

The procedural details and outcomes until discharge was complete for 100% of patients. Patients received LAAC because of prior bleeding (66%), thromboembolism (10.0%) or other reasons (24%). An isolated LAAC procedure was planned in 90.0% of patients; combined procedures included closure of the patent foramen ovale or atrial septal defect (5%), cardiac ablation (3%), edge-to-edge mitral repair (2%), coronary angiography (1%), pacemaker implantation (0.2%), and transcutaneous aortic valve implantation (0.2%). The most common anticipated device was Amplatzer (56%), followed by Watchman (43%), and LAmbre (0.5%).

The median procedural duration was 70 (IQR 53–98) minutes with a median fluoroscopy time of 16 (11–22) minutes and a dose area product of 4433 µGym^2^ (Table 2). The device was successfully implanted in 97% of the cases.

A complication occurred in 19% of patients and major procedural complications arose in 7%. Access site complications requiring intervention occurred in 3%, cardiac tamponade requiring intervention in 2%, and ischaemic stroke in 1% of patients (Table 2). Furthermore, the following complications occurred in the whole population: cardiopulmonary resuscitation (0.5%, *n* = 2), open heart surgery (surgical retrieval of an embolized and dislocated device and surgical treatment of cardiac perforation with tamponade; 0.5%, *n* = 2), and interventional retrieval of dislocated LAAC device (0.5%, *n* = 2; Table 2).

Periprocedural survival was 99.5%. Two deaths in association with the procedure occurred:One patient had dislocation of the LAAC device into the left atrium two days after the procedure, requiring cardiac surgery. The patient aspirated in the postoperative phase, developed a severe pneumonia, and died due to respiratory failure at day 27 after LAAC.One patient developed massive throat bleeding, probably caused by the insertion of the transoesophageal probe, leading to tracheal obstruction on the day of the LAAC. Despite an emergency tracheotomy and resuscitation, the patient deteriorated and passed away in the operating theatre.

Remaining patients were discharged after a median of 2 (IQR, 1–3) days.

Minor complications occurred in 16.8% of patients, including a transfer to the intensive care unit, catecholamine support (6% each), failure to implant the device (3%), bleeding requiring transfusion (3%), new pericardial effusion without intervention (5%), and prolonged hospital stay (4%).

### 3.3. Predictors for Complications

End-stage kidney injury requiring chronic dialysis was associated with major complications with a bivariate OR of 16 (95% CI 3–70, *p* = 0.001). The prevalence of end-stage kidney disease among patients with and without complications was 14% and 1%, respectively. Furthermore, low haemoglobin and low erythrocytes were associated with increased risk of complications (haemoglobin: 11.3 vs. 12.7 mg/dL in patients with vs. without complications, OR 0.8, 95% 0.6–0.9, *p* = 0.010; haematocrit: 35% vs. 38%, *p* = 0.016). Anaemic patients had a 2.2-fold increased risk of major complications (9.6% vs. 4.4%, *p* = 0.047).

There were no differences in major complications among patients receiving Amplatzer vs. Watchman devices (8% vs. 5%, *n* = 0.240). However, minor complications occurred significantly more often in the Amplatzer group (22% vs. 9%, *p* < 0.001), mostly due to a higher rate of shock (9% vs. 1%, *p* < 0.001). Other distinct complications were similar between both devices. Most notably, procedure duration (83 vs. 64 min, *p* = 0.001) and dose area product (6972 vs. 1750 µGym^2^, *p* < 0.001) were increased in patients receiving Amplatzer devices.

Neither minor complications, major complications, nor periprocedural death were correlated with low centre volume (*p* = 0.107 for major complications, Figure 1). The major complication rate was similar between low-volume centres (defined as total procedure count below the median, 3%) and high-volume centres (remaining centres, 8%, *p* = 0.268).

### 3.4. Multivariate Analysis

When adjusted for age and left ventricular function, both dialysis (OR 13, 95% confidence interval 2.5–68.5, *p* = 0.002) and haemoglobin (OR 0.80, 95% CI 0.65–0.99 per g/dL, *p* = 0.040) remained as independent predictors of major complications and death (Table 3).

### 3.5. Complications and Long-Term Outcomes

One-year follow up regarding mortality was complete in 78% of patients. Major complications were associated with reduced one-year survival (cumulative survival 85.0% vs. 94.2%, *p* = 0.040, Figure 2).

## 4. Discussion

This study shows that (1) major complications and procedural death occurred relatively rarely in a high-risk patient cohort receiving LAAC, (2) end-stage kidney failure and low haemoglobin were independently associated with a worse short-term outcome after LAAC, and (3) low centre experience was not associated with increased major complication rates, which themselves (4) were associated with higher long-term mortality.

Currently, patients undergoing LAAC in Austria represent a high-risk cohort with a mean CHA_2_DS_2_-VASc score of 4.4, a median HAS-BLED score of 3.1, and a history of stroke in almost half of patients (47%). Due to limitations in reimbursement, a dramatic ischaemic or haemorrhagic event usually precedes the decision to perform LAAC in most patients [11]. Baseline ischaemic and haemorrhagic risk is similar to the Left Atrial Appendage Closure Registry [4].

As LAAC is a preventive measure without any acute symptom improvement, any complication imposes a burden on a patient who otherwise would have been asymptomatic. Furthermore, the beneficial effect of LAAC cannot be predicted on an individual level. Therefore, the decision to perform LAAC must be evaluated thoroughly, and the complication rate must be kept at a low level. This is especially important as major complications were associated with long-term mortality in this cohort.

### 4.1. Types of Complications

The most common major complications are access site complications. The use of ultrasound-guided puncture and adequate post-procedural management must be emphasized, as well as vascular closure devices as appropriate [5].

Pericardial tamponade occurs in a substantial proportion of patients and requires quick action from the medical team. Fortunately, the prognosis after interventional pericardiocentesis is good. In this analysis, there was only one case requiring bailout heart surgery with a positive result. Emergency plans and standard operation procedures for handling acute cardiac tamponade are required to be in place, with fast transfer to a cardiac surgery department in the event of a failed interventional drainage.

Furthermore, the device dislocated in three cases, but interventional capture was possible in two of them. Unfortunately, ischaemic stroke also occurred in a proportion of patients during the procedure.

The reduction of periprocedural complications may be achieved through adequate patient selection, patient preparation, and optimized procedural management. The preprocedural workflow may be further supported by individual computational simulation technology [14].

Fortunately, “typical” complications of LAAC were treated very well without immediate mortality. Only one patient suffered from throat bleeding, a rare complication of LAAC, and died in the cath lab. Further deaths may have been prevented by adequate training of the responsible medical staff for common complications.

### 4.2. Predictors for Complications

Currently, there is no optimal treatment strategy for stroke prevention in patients on dialysis, as they have both a high ischaemic and bleeding risk, and the experience of DOACs in this patient population is limited [1]. On the one hand, these patients have a ten times elevated risk of stroke [15] and may therefore benefit most from LAAC. However, end-stage renal failure is also associated with competing complications that lead to morbidity and mortality, such as acute myocardial infarction, heart failure, and sudden cardiac death [16]. In this analysis, end-stage chronic kidney failure was independently associated with increased major complication rates. Therefore, the benefits and risks of LAAC have to be weighed carefully in this fragile patient population. These results are in conflict with Jamal et al. and Benini Tapias et al., who did not find a worse outcome in LAAC patients with chronic kidney disease [17,18]. In other cardiac interventions, such as percutaneous coronary intervention and transcatheter aortic valve implantation, end-stage renal disease was also associated with poorer acute outcomes [19,20].

A low haemoglobin level was independently associated with increased major complications in the multivariable analysis, although periprocedural bleeding requiring transfusion was not defined as a major complication. Of note, there was no association between the history of anaemia, previous bleeding, or previous blood transfusions and major complications. We hypothesise that suboptimal preparation before the procedure, for example, hypoferremia or uncorrected anaemia after a recent bleeding event, may be associated with increased complications. Patients with low haemoglobin levels undergoing LAAC may have fewer physiological resources to cope with acute stress on the cardiovascular system, leading to quicker deterioration and therefore a higher risk of complications. It can be speculated that appropriate preparation, such as control of the bleeding and iron supplementation, may have ameliorated short-term outcomes. While low haemoglobin levels were associated with poorer outcomes in elective non-cardiac surgery [21,22] and cardiac surgery [23], this is the first study to evaluate haemoglobin on acute outcome in LAAC. Despite low evidence from mainly retrospective analyses [24], there is general agreement that adequate preoperative preparation may be useful in patients with anaemia [22]. Laboratory parameters other than haemoglobin and haematocrit were not associated with the periprocedural complication rate.

Regarding procedural management, we did not find differences in major complications in patients receiving Amplatzer and Watchman devices. This is reassuring, as there was a significant difference described by Qiao et al. and Lakkireddy et al. [25,26]. However, the rate of minor complications was higher in Amplatzer devices. It is unclear to the authors if this difference is relevant as the administration of catecholamines in patients with low blood pressure and the threshold for transfer to the intensive care unit may vary between centres. Of note, minor complications were not spread equally across centres (Figure 1).

In our analysis, there was no sign of increased complications in centres with low experience (Figure 1). The major complication rate was 3% in low-volume centres and 8% in high-volume centres (*p* = ns). A previous report from 2016 showed better outcomes and fewer complications after LAAC of 30 patients [27]. Proper proctoring and optimization of standard operating procedures may have reduced the risk of early procedural complications in learning investigators.

Furthermore, literature suggests liver cirrhosis [28], female sex [29], and increased age combined with previous gastrointestinal bleeding [30] are predictive factors for procedural complications. These parameters were not associated with a worse prognosis in this analysis.

### 4.3. Limitations

While the major strengths of this study are the inclusion of all consecutive patients undergoing LAAC in a whole country, external evaluation of clinical data, and complete follow up regarding survival, it still represents a retrospective study with all forms of associated bias. Furthermore, as this cohort presents a high-risk cohort, the results may not be applicable to other patient cohorts with different risk profiles. Finally, this analysis focused on procedural complications and not on the overall long-term benefits of LAAC.

## 5. Conclusions

Despite the fact that the patient cohort undergoing LAAC in Austria resembled a high-risk cohort, the rate of major procedural complications was considerably low. Chronic dialysis and low haemoglobin were independent predictors of complications.

## Figures and Tables

**Figure 1 jcm-11-06548-f001:**
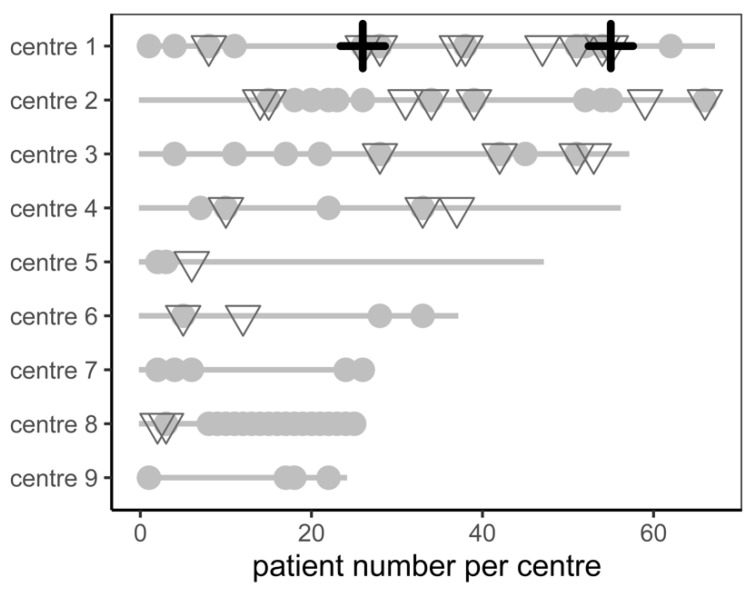
Patient count, minor complications (grey dot), major complications (grey triangle), and periprocedural mortality (black cross) stratified by centre.

**Figure 2 jcm-11-06548-f002:**
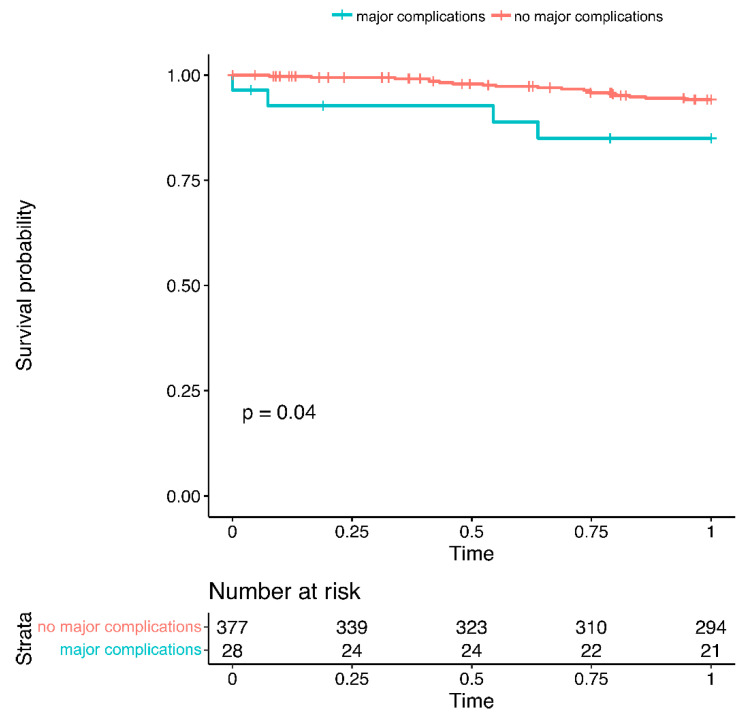
One-year mortality in patients with vs. without major complications.

**Table 1 jcm-11-06548-t001:** Baseline characteristics.

Parameter	Total Population	Patients with Complications (*n* = 28)	Patients without Complications (*n* = 377)	*p* Value
**Baseline Demographics**				
female gender	36.5% (*n* = 148)	50% (*n* = 14)	35.5% (*n* = 134)	0.154
age (years)	75 (70–79)	75.5 (72–79)	75 (70–79)	0.679
body mass index (kg/m^2^)	27 (24–30)	29 (25–32)	27 (24–30)	0.424
body surface area (m^2^)	1.92 ± 0.21	1.88 ± 0.20	1.92 ± 0.21	0.316
**Comorbidities**				
CHA_2_DS_2_-VASc score	4 (3–5)	5 (4–6)	4 (3–5)	0.408
HAS-BLED score	3 (2–4)	3 (3–4)	3 (2–4)	0.797
arterial hypertension	88.4% (*n* = 358)	89.3% (*n* = 25)	88.3% (*n* = 333)	1.000
diabetes mellitus	28.1% (*n* = 114)	32.1% (*n* = 9)	27.9% (*n* = 105)	0.664
congestive heart failure	24% (*n* = 97)	17.9% (*n* = 5)	24.4% (*n* = 92)	0.501
stroke	46.7% (*n* = 189)	53.6% (*n* = 15)	46.2% (*n* = 174)	0.557
ischaemic stroke	25.2% (*n* = 102)	32.1% (*n* = 9)	24.7% (*n* = 93)	0.373
haemorrhagic stroke intracerebral bleeding subarachnoid bleeding subdural bleeding epidural bleeding	25.9% (*n* = 105) 22% (*n* = 89) 4.2% (*n* = 17) 4.7% (*n* = 19) 1.2% (*n* = 5)	25% (*n* = 7) 17.9% (*n* = 5) 10.7% (*n* = 3) 3.6% (*n* = 1) 0% (*n* = 0)	26% (*n* = 98) 22.3% (*n* = 84) 3.7% (*n* = 14) 4.8% (*n* = 18) 1.3% (*n* = 5)	1.000 0.813 0.105 1.000 1.000
chronic kidney disease	19.5% (*n* = 79)	25% (*n* = 7)	19.1% (*n* = 72)	0.459
chronic liver disease	5.4% (*n* = 22)	10.7% (*n* = 3)	5% (*n* = 19)	0.188
anaemia	38.0% (*n* = 154)	32.1% (*n* = 9)	38.5% (*n* = 145)	0.552
prior blood transfusion	29.4% (*n* = 119)	21.4% (*n* = 6)	30% (*n* = 113)	0.396
coronary artery disease	41.2% (*n* = 167)	35.7% (*n* = 10)	41.6% (*n* = 157)	0.691
cerebral artery disease	13.8% (*n* = 56)	17.9% (*n* = 5)	13.5% (*n* = 51)	0.568
periphery artery disease	9.6% (*n* = 39)	7.1% (*n* = 2)	9.8% (*n* = 37)	1.000
chronic obstructive pulmonary disease	12.8% (*n* = 52)	10.7% (*n* = 3)	13% (*n* = 49)	1.000
chronic dialysis	2.0% (*n* = 8)	14.3% (*n* = 4)	1.1% (*n* = 4)	0.001
paroxysmal AF	34.1% (*n* = 138)	46.4% (*n* = 13)	33.2% (*n* = 125)	0.155
vascular malformation cerebral upper gastrointestinal tract lower gastrointestinal tract	5.2% (*n* = 21) 4.2% (*n* = 17) 5.2% (*n* = 21)	10.7% (*n* = 3) 10.7% (*n* = 3) 0.0% (*n* = 0)	4.8% (*n* = 18) 3.7% (*n* = 14) 5.6% (*n* = 21)	0.170 0.105 0.383
prior acute coronary syndrome	13.3% (*n* = 54)	14.3% (*n* = 4)	13.3% (*n* = 50)	0.778
prior pulmonary embolism	1.7% (*n* = 7)	0.0% (*n* = 0)	1.9% (*n* = 7)	1.000
prior peripheral embolism	2.0% (*n* = 8)	0.0% (*n* = 0)	2.1% (*n* = 8)	1.000
**Echocardiography**				
LVEF normal 35–50% <35%	74.6% (*n* = 302) 18.0% (*n* = 73) 7.4% (*n* = 30)	75.0% (*n* = 21) 21.4% (*n* = 6) 3.6% (*n* = 1)	74.5% (*n* = 281) 17.8% (*n* = 67) 7.7% (*n* = 29)	0.786
severe aortic stenosis	1.2% (*n* = 5)	3.6% (*n* = 1)	1.1% (*n* = 4)	0.302
severe mitral regurgitation	6.2% (*n* = 25)	14.3% (*n* = 4)	5.6% (*n* = 21)	0.084
severe tricuspid regurgitation	4.9% (*n* = 20)	3.6% (*n* = 1)	5% (*n* = 19)	1.000
**Laboratory**				
erythrocytes (T/L)	4.3 ± 0.67	4.1 ± 0.84	4.32 ± 0.65	0.216
haemoglobin (g/dL)	12.5 (11.0–14.1)	11.3 (10.4–12.9)	12.7 (11.0–14.2)	0.010
haematocrit (%)	38 (33–42)	35 (32–38)	38 (33–42)	0.016
platelets (G/L)	219 (173–261)	191 (173–253)	219 (174–262.5)	0.289
NT-ProBNP (ng/L)	912 (382–2153)	1611 (410–2713)	885 (384–2063)	0.339
creatinine (mg/dL)	1.10 (0.90–1.41)	1.10 (0.86–1.67)	1.10 (0.9–1.4)	0.892
eGFR (ml/min/1.73 m^2^)	75.0 (67.9–81.9)	74.5 (63.6–82.3)	75.1 (68.0–81.7)	0.559
ASAT (U/L)	24 (20–30)	23 (19–29)	24 (20–30)	0.745
ALAT (U/L)	20 (15–29)	16 (13–26)	20 (15–29)	0.073
INR	1.1 (1–1.2)	1.1 (1–1.11)	1.1 (1–1.2)	0.703
aPTT (sec)	34 (30–40)	34 (32–37.25)	34 (30–40)	0.754
albumine (g/L)	41 (36–44)	42 (32–44)	41 (36–44)	0.544
total protein (g/L)	70 (65–74)	66 (58.5–72)	71 (66–74)	0.055
total cholesterol (mg/dL)	152 (123–189)	156 (138–201)	152 (123–189)	0.511
LDL (mg/dL)	87 (65–113)	94 (72–126)	87 (65–112)	0.358
triglycerides (mg/dL)	98 (73–144)	103 (72–160)	98 (74–143)	0.657
**Indication for LAAC**				
indication group bleeding other thromboembolism	66.2% (*n* = 268) 24.2% (*n* = 98) 9.6% (*n* = 39)	57.1% (*n* = 16) 28.6% (*n* = 8) 14.3% (*n* = 4)	66.8% (*n* = 252) 23.9% (*n* = 90) 9.3% (*n* = 35)	0.435
primary indication for LAAC gastrointestinal bleeding intracranial Bleeding bleeding under OAC stroke other predisposition to bleeding other contraindication to OAC embolism despite OAC anaemia OAC intolerance patient preference requirement for triple therapy epistaxis	29.1% (*n* = 118) 28.4% (*n* = 115) 6.9% (*n* = 28) 5.7% (*n* = 23) 4.7% (*n* = 19) 4.4% (*n* = 18) 4.2% (*n* = 17) 4.0% (*n* = 16) 3.7% (*n* = 15) 3.2% (*n* = 13) 2.0% (*n* = 8) 2.0% (*n* = 8) 1.7% (*n* = 7)	21.4% (*n* = 6) 32.1% (*n* = 9) 3.6% (*n* = 1) 10.7% (*n* = 3) 0.0% (*n* = 0) 7.1% (*n* = 2) 7.1% (*n* = 2) 3.6% (*n* = 1) 3.6% (*n* = 1) 3.6% (*n* = 1) 7.1% (*n* = 2) 0.0% (*n* = 0) 0.0% (*n* = 0)	29.7% (*n* = 112) 28.1% (*n* = 106) 7.2% (*n* = 27) 5.3% (*n* = 20) 5.0% (*n* = 19) 4.2% (*n* = 16) 4.0% (*n* = 15) 4.0% (*n* = 15) 3.7% (*n* = 14) 3.2% (*n* = 12) 1.6% (*n* = 6) 2.1% (*n* = 8) 1.9% (*n* = 7)	0.588
contraindication for OAC	20.7% (*n* = 84)	32.1% (*n* = 9)	19.9% (*n* = 75)	0.146
device Amplatzer Watchman LAmbre	56.3% (*n* = 228) 43.2% (*n* = 175) 0.5% (*n* = 2)	67.9% (*n* = 19) 32.1% (*n* = 9) 0.0% (*n* = 0)	55.4% (*n* = 209) 44% (*n* = 166) 0.5% (*n* = 2)	0.342
isolated procedure	89.6% (*n* = 363)	78.6% (*n* = 22)	90.5% (*n* = 341)	0.057

eGFR: estimated glomerular filtration rate; LAAC: left atrial appendage closure; LDL: low-density lipoprotein; LVEF: left ventricular ejection fraction; OAC: oral anticoagulation.

**Table 2 jcm-11-06548-t002:** Procedural outcome.

Parameter	Total Population
fluoroscopy time (min)	16 (11–23)
dose area product (µGym^2^)	4430 (1574–8991)
contrast medium (mL)	100 (68–139)
procedure duration (min)	70 (53–98)
any complication	19.3% (*n* = 78)
**Major Complications**	6.9% (*n* = 28)
pericardial tamponade	2.2% (*n* = 9)
access site complications requiring surgery requiring thrombin injection requiring angioseal	3.2% (*n* = 11) 2.0% (*n* = 8) 0.7% (*n* = 3) 0.5% (*n* = 2)
stroke	1.2% (*n* = 5)
death	0.5% (*n* = 2)
heart surgery	0.5% (*n* = 2)
interventional retrieval of dislocated LAAC device	0.5% (*n* = 2)
cardiopulmonary resuscitation	0.5% (*n* = 2)
device embolization	0.2% (*n* = 1)
**Minor Complications**	16.8% (*n* = 68)
admission to intensive care	5.7% (*n* = 23)
shock	5.7% (*n* = 23)
device not implanted	3.2% (*n* = 13)
bleeding requiring transfusion	2.5% (*n* = 10)
new pericardial effusion (no therapy)	4.7% (*n* = 19)
iatrogenic atrial septum defect	0.5% (*n* = 2)
prolonged hospital stay (>14 days)	3.5% (*n* = 14)

**Table 3 jcm-11-06548-t003:** Multivariate logistic regression analysis with the endpoint as major periprocedural complications.

Parameter	Bivariate Analysis		Multivariable Analysis	
	OR (95% CI)	*p* Value	OR (95% CI)	*p* Value
dialysis	15.54 (3.49–69.50)	0.001	13.0 (2.5–68.5)	0.002
haemoglobin (per g/dL)	0.78 (0.63–0.94)	0.010	0.80 (0.65–0.99)	0.040
age (per year)	1.00 (0.96–1.05)	0.679	0.42 (0.05–3.66)	0.431
LVEF < 35%	0.44 (0.02–2.21)	0.710	1.01 (0.95–1.06)	0.847
LVEF 35–50%	1.26 (0.45–3.06)	0.613	1.04 (0.38–2.82)	0.945

## Data Availability

Raw data are available upon reasonable request from the corresponding author.

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
