# Peer review of "Not to Rush—Laboratory Parameters and Procedural Complications in Patients Undergoing Left Atrial Appendage Closure"

_jcm, 2022, doi:10.3390/jcm11216548_

Round 1
Reviewer 1 Report
Colleagues Zweiker et al. report in their manuscript, entitled "Not to rush – Laboratory parameters and procedural complica- 2 tions in patients undergoing left atrial appendage closure" about LAAC and determing risk factors for outcome.
They found low haemoglobin and end-stage kidney disease to be independent risk factors. They conclude, that also low centre experience is associated with higher long-term mortality, while the procedure itself is at low procedural risk.
The manuscript is well written, nevertheless, I have some comments, that need to be addressed.
3.4. Multivariate analyis
You state, that dialysis was an independent risk factor for outcome. As these patients have a high mortality per se, and might probably benefit from stroke prevention more than other patients, such element of discussion is missing in the discussion section completely. Also intermittend anticoagulation during dialysis might have a "protective" effect on stroke. Please discuss and add references.
3.5. Complications and long-term outcome
A major limitation is the low complete follow-up of 78%. Could you enhance your efforts to complete follow-up? Maybe a shorter follow-up could lead to a higher rate of follow-up.
4.2. Predictors for complications:
"We hypothesize that suboptimal prep- aration before the procedure, for example hypoferremia or uncorrected anaemia after a recent bleeding event may be associated with increased complications"
Why that? It would make more sense, if bad baseline conditions lead to worse outcome, especially when complications occur. Why shoud e.g. low haemoglobin lead to more complications, if you mean procedural complications. Please clarify, especially as you write, that transfusion was not defined as a major complication.
Discussion
You should extend your discussion on your main findings: preprocedural labarotories, multivariate analysis, center experience. Is there a critical number of procedures at a center, that lead to a better outcome?
Author Response
3.4. Multivariate analyis
You state, that dialysis was an independent risk factor for outcome. As these patients have a high mortality per se, and might probably benefit from stroke prevention more than other patients, such element of discussion is missing in the discussion section completely. Also intermittend anticoagulation during dialysis might have a "protective" effect on stroke. Please discuss and add references.
- We thank the reviewer for this comment. The fact that patients on dialysis have a high rate of mortality and morbidity is very important. Therefore, they will have a higher risk of stroke than the remaining population and therefore, LAAC will be more beneficial on the long run. However, this is a two-sided sword. As overall mortality is higher, they may not have that much time to benefit from the protective effect of LAAC until other complications, such as sudden cardiac death, myocardial infarction or heart failure occur. We added these comments to the discussion at section 4.2 (lines 248-253).
3.5. Complications and long-term outcome
A major limitation is the low complete follow-up of 78%. Could you enhance your efforts to complete follow-up? Maybe a shorter follow-up could lead to a higher rate of follow-up.
As this registry includes all patients receiving LAAC from 9 centres in Austria, there is a lot of diversity regarding follow up procedures. For this analysis, we only used clinical data available to the implanting centres. Unfortunately, a number of patients was not followed by the implanting centre but at smaller centres. To reduce bias by missing follow up, we concentrated on in-hospital complications, which were available in 100% of patients. We added information about complete in-hospital follow up at the beginning of section 3.2. (line 136).4.2. Predictors for complications:
"We hypothesize that suboptimal preparation before the procedure, for example hypoferremia or uncorrected anaemia after a recent bleeding event may be associated with increased complications". Why that? It would make more sense, if bad baseline conditions lead to worse outcome, especially when complications occur. Why shoud e.g. low haemoglobin lead to more complications, if you mean procedural complications. Please clarify, especially as you write, that transfusion was not defined as a major complication.
- Our thoughts were as follows: The main indication for LAAC remains bleeding, especially gastrointestinal or cerebral. Therefore, a high number of patients has a recent history of bleeding. These patients may have a low haemoglobin at baseline and therefore a worse outcome after LAAC. We hypothesize that those patients may have a better outcome if the anaemia was corrected before the procedure by both adequate control of bleeding and iron supplementation.
- Interestingly, during investigation of existing literature we found that low haemoglobin levels were indeed associated with reduced outcome, not limited to bleeding, in other patient populations receiving non-cardiac surgery (doi: doi:10.1111/anae.13840 and 10.1001/jama.2019.0554) and cardiac surgery (doi: 10.2215/cjn.00110114). In the 2018 Frankfurt Consensus Conference regarding patient blood management, the authors recommend the “Use of iron supplementation to reduce red blood cell transfusion rate in adult preoperative patients with iron-deficient anaemia undergoing elective surgery” to reduce the rate of blood transfusions and other perioperative complications.
- We already mentioned in the discussion that “periprocedural bleeding requiring transfusion was not defined as major complication” (line 262), but we added the sentence “Bleeding requiring transfusion was explicitly defined as minor procedural complication.” in the methods section (lines 103-104), as well as the sentence “It can be speculated that appropriate preparation, such as control of the bleeding and iron supplementation, may have ameliorated short-term outcome.” in the discussion (269-271).
Discussion
You should extend your discussion on your main findings: preprocedural labarotories, multivariate analysis, center experience. Is there a critical number of procedures at a center, that lead to a better outcome?
- We thank the reviewer for this comment. We substantially adapted our discussion to accommodate all mentioned findings: preprocedural laboratories other than haemoglobin/haematocrit (lines 275-276), multivariate analysis (lines 253 and 261-262), center experience (lines 285-289). As there was no difference in complications between low and high volume centres, we did not find a critical number of procedures that would lead to a better outcome, in contrast to previous analyses (doi: 10.1111/joic.12316). We hypothesize that proctoring may have attenuated the learning curve (as discussed at lines 288-290).
Reviewer 2 Report
In the current study by Zweiker et al. the authors aim to predict major complications in a retrospective cohort of high-risk patients undergoing left atrial appendage closure.
The paper is well written, statistical analysis were performed in a univariable and multivariable matter. Therefore, the data appear to be robust.
As predictive factors for major compilations only chronic kidney disease requiring hemodialysis and anemia could be detected. The reasons why these two factors are associated with higher complications rates remain elusive. However, end stage renal disease is a common risk factor for poor outcome in patients undergoing cardiac interventions. Therefore, clinical impact of this findings remains low.
In summary, this study shows a relatively low rate of complications even in a cohort of high-risk patients (according to CHA2DS2-VASc-Score). This shows, that interventional LAAC is save and provides an alternative tool for stroke prevention in high risk patients requiring oral anticoagulation.
Major:
Please comment on the risk for major complications in chronic kidney disease patients with or without hemodialysis in also cohorts of patients undergoing other cardiac interventions.
Minor:
Line 43 “forty-seven percent”
à please use numbers and percent consistently
Line 54 “non-medicinal” à interventional
Author Response
Major:
Please comment on the risk for major complications in chronic kidney disease patients with or without hemodialysis in also cohorts of patients undergoing other cardiac interventions.
- We performed another literature search and found end-stage kidney disease to be associated with worse acute outcome after both TAVI and PCI. We added a sentence referencing to the evidence at section 4.2. (lines 257-260).
Minor:
Line 43 “forty-seven percent” - please use numbers and percent consistently
- We thank the reviewer for this comment and changed it to “47%”.
Line 54 “non-medicinal” à interventional
- We agree that “interventional” is indeed a better word than “non-medicinal”. We changed the sentence appropriately.
Reviewer 3 Report
The study is well written . The authors found that the complications rate of high risk patients is low. Chronic dialysis and low Hb are independent risk factors to short term complications.
'
- Do you have data regarding efficacy ?? It is important to know that the complication rate is low and know the predictors of complications, but we need to know whom patients will get benefit,. We need to know the benefit risk profile.
_ The authors should discuss in details if dialysis patients should or should not recieve LAAC.
Author Response
Do you have data regarding efficacy ?? It is important to know that the complication rate is low and know the predictors of complications, but we need to know whom patients will get benefit,. We need to know the benefit risk profile.
- The authors agree with this statement. However, this analysis focused on procedural complications and not on the overall long-term benefit, due to limited follow up (78% at one year). We added this limitation to the discussion at lines 300-302.
The authors should discuss in details if dialysis patients should or should not recieve LAAC.
- This is indeed a very difficult question. Dialysis patients have a higher risk of stroke and therefore may benefit more from LAAC. On the other hand, they also have a higher risk of death and other complications, such as heart failure and sudden cardiac death. They therefore may not have enough time to benefit from the procedure. We added these comments to the discussion at lines 248-252.
Round 2
Reviewer 1 Report
Accept in the corrected form, all comments have been addressed.
Reviewer 3 Report
The authors answered all comments